# Optimal Algorithms for Non-Smooth Distributed Optimization in Networks

**Kevin Scaman**[1] **Francis Bach**[2] **Sébastien Bubeck**[3] **Yin Tat Lee**[3,4] **Laurent Massoulié**[2,5]

[1] Huawei Noah's Ark Lab, [2] INRIA, Ecole Normale Supérieure, PSL Research University,
[3] Microsoft Research, [4] University of Washington, [5] MSR-INRIA Joint Centre

## Abstract

In this work, we consider the distributed optimization of non-smooth convex functions using a network of computing units. We investigate this problem under two regularity assumptions: (1) the Lipschitz continuity of the *global* objective function, and (2) the Lipschitz continuity of *local* individual functions. Under the *local regularity* assumption, we provide the first optimal first-order decentralized algorithm called *multi-step primal-dual* (MSPD) and its corresponding optimal convergence rate. A notable aspect of this result is that, for non-smooth functions, while the dominant term of the error is in $O(1/\sqrt{t})$, the structure of the communication network only impacts a second-order term in $O(1/t)$, where $t$ is time. In other words, the error due to limits in communication resources decreases at a fast rate even in the case of non-strongly-convex objective functions. Under the *global regularity* assumption, we provide a simple yet efficient algorithm called *distributed randomized smoothing* (DRS) based on a local smoothing of the objective function, and show that DRS is within a $d^{1/4}$ multiplicative factor of the optimal convergence rate, where $d$ is the underlying dimension.

## 1 Introduction

Distributed optimization finds many applications in machine learning, for example when the dataset is large and training is achieved using a cluster of computing units. As a result, many algorithms were recently introduced to minimize the average $\bar{f} = \frac{1}{n} \sum_{i=1}^{n} f_i$ of functions $f_i$ which are respectively accessible by separate nodes in a network [1, 2, 3, 4]. Most often, these algorithms alternate between local and incremental improvement steps (such as gradient steps) with communication steps between nodes in the network, and come with a variety of convergence rates (see for example [5, 4, 6, 7]).

Recently, a theoretical analysis of first-order distributed methods provided optimal convergence rates for strongly-convex and smooth optimization in networks [8]. In this paper, we extend this analysis to the more challenging case of non-smooth convex optimization. The main contribution of this paper is to provide optimal convergence rates and their corresponding optimal algorithms for this class of distributed problems under two regularity assumptions: (1) the Lipschitz continuity of the *global* objective function $\bar{f}$, and (2) a bound on the average of Lipschitz constants of *local* functions $f_i$.

Under the *local regularity* assumption, we provide in Section 4 matching upper and lower bounds of complexity in a decentralized setting in which communication is performed using the *gossip* algorithm [9]. Moreover, we propose the first optimal algorithm for non-smooth decentralized optimization, called *multi-step primal-dual* (MSPD). Under the more challenging *global regularity* assumption, we show in Section 3 that distributing the simple smoothing approach introduced in [10] yields fast convergence rates with respect to communication. This algorithm, called *distributed randomized smoothing* (DRS), achieves a convergence rate matching the lower bound up to a $d^{1/4}$ multiplicative factor, where $d$ is the dimensionality of the problem.

Our analysis differs from the smooth and strongly-convex setting in two major aspects: (1) the naïve *master/slave* distributed algorithm is in this case not optimal, and (2) the convergence rates differ between communication and local computations. More specifically, error due to limits in communication resources enjoys fast convergence rates, as we establish by formulating the optimization problem as a composite saddle-point problem with a smooth term for communication and non-smooth term for the optimization of the local functions (see Section 4 and Eq. (21) for more details).

**Related work.** Many algorithms were proposed to solve the decentralized optimization of an average of functions (see for example [1, 11, 3, 4, 12, 2, 13, 5]), and a sheer amount of work was devoted to improving the convergence rate of these algorithms [5, 6]. In the case of non-smooth optimization, fast communication schemes were developed in [14, 15], although precise optimal convergence rates were not obtained. Our decentralized algorithm is closely related to the recent primal-dual algorithm of [14] which enjoys fast communication rates in a decentralized and stochastic setting. Unfortunately, their algorithm lacks gossip acceleration to reach optimality with respect to communication time. Finally, optimal convergence rates for distributed algorithms were investigated in [8] for smooth and strongly-convex objective functions, and [16, 17] for totally connected networks.

## 2 Distributed optimization setting

**Optimization problem.** Let $\mathcal{G} = (\mathcal{V}, \mathcal{E})$ be a strongly connected directed graph of $n$ computing units and diameter $\Delta$, each having access to a convex function $f_i$ over a convex set $\mathcal{K} \subset \mathbb{R}^d$. We consider minimizing the average of the local functions

$$\min_{\theta \in \mathcal{K}} \bar{f}(\theta) = \frac{1}{n} \sum_{i=1}^{n} f_i(\theta), \tag{1}$$

in a distributed setting. More specifically, we assume that each computing unit can compute a subgradient $\nabla f_i(\theta)$ of its own function in one unit of time, and communicate values (i.e. vectors in $\mathbb{R}^d$) to its neighbors in $\mathcal{G}$. A *direct* communication along the edge $(i, j) \in \mathcal{E}$ requires a time $\tau \geq 0$. These actions may be performed asynchronously and in parallel, and each machine $i$ possesses a local version of the parameter, which we refer to as $\theta_i \in \mathcal{K}$.

**Regularity assumptions.** Optimal convergence rates depend on the precise set of assumptions applied to the objective function. In our case, we will consider two different constraints on the regularity of the functions:

(A1) **Global regularity:** the (global) function $\bar{f}$ is convex and $L_g$-Lipschitz continuous, in the sense that, for all $\theta, \theta' \in \mathcal{K}$,

$$|\bar{f}(\theta) - \bar{f}(\theta')| \leq L_g \|\theta - \theta'\|_2. \tag{2}$$

(A2) **Local regularity:** Each local function is convex and $L_i$-Lipschitz continuous, and we denote as $L_\ell = \sqrt{\frac{1}{n} \sum_{i=1}^{n} L_i^2}$ the $\ell_2$-average of the local Lipschitz constants.

Assumption (A1) is *weaker* than (A2), as we always have $L_g \leq L_\ell$. Moreover, we may have $L_g = 0$ and $L_\ell$ arbitrarily large, for example with two linear functions $f_1(x) = -f_2(x) = ax$ and $a \to +\infty$. We will see in the following sections that the local regularity assumption is easier to analyze and leads to matching upper and lower bounds. For the global regularity assumption, we only provide an algorithm with a $d^{1/4}$ competitive ratio, where $d$ is the dimension of the problem. Finding an optimal distributed algorithm for global regularity is, to our understanding, a much more challenging task and is left for future work.

Finally, we assume that the feasible region $\mathcal{K}$ is convex and bounded, and denote by $R$ the radius of a ball containing $\mathcal{K}$, i.e.

$$\forall \theta \in \mathcal{K}, \ \|\theta - \theta_0\|_2 \leq R, \tag{3}$$

where $\theta_0 \in \mathcal{K}$ is the initial value of the algorithm, that we set to $\theta_0 = 0$ without loss of generality.

**Black-box optimization procedure.** The lower complexity bounds in Theorem 2 and Theorem 3 depend on the notion of black-box optimization procedures of [8] that we now recall. A black-box optimization procedure is a distributed algorithm verifying the following constraints:

1. **Local memory:** each node $i$ can store past values in a (finite) internal memory $\mathcal{M}_{i,t} \subset \mathbb{R}^d$ at time $t \geq 0$. These values can be accessed and used at time $t$ by the algorithm run by node $i$, and are updated either by local computation or by communication (defined below), that is, for all $i \in \{1, ..., n\}$,

$$\mathcal{M}_{i,t} \subset \mathcal{M}_{i,t}^{comp} \cup \mathcal{M}_{i,t}^{comm}. \tag{4}$$

2. **Local computation:** each node $i$ can, at time $t$, compute a subgradient of its local function $\nabla f_i(\theta)$ for a value $\theta \in \mathcal{M}_{i,t-1}$ in the node's internal memory before the computation.

$$\mathcal{M}_{i,t}^{comp} = \mathrm{Span}\left(\{\theta, \nabla f_i(\theta) : \theta \in \mathcal{M}_{i,t-1}\}\right). \tag{5}$$

3. **Local communication:** each node $i$ can, at time $t$, share a value to all or part of its neighbors, that is, for all $i \in \{1, ..., n\}$,

$$\mathcal{M}_{i,t}^{comm} = \mathrm{Span}\left(\bigcup_{(j,i) \in \mathcal{E}} \mathcal{M}_{j,t-\tau}\right). \tag{6}$$

4. **Output value:** each node $i$ must, at time $t$, specify one vector in its memory as local output of the algorithm, that is, for all $i \in \{1, ..., n\}$,

$$\theta_{i,t} \in \mathcal{M}_{i,t}. \tag{7}$$

Hence, a black-box procedure will return $n$ output values—one for each computing unit—and our analysis will focus on ensuring that *all local output values* are converging to the optimal parameter of Eq. (1). For simplicity, we assume that all nodes start with the simple internal memory $\mathcal{M}_{i,0} = \{0\}$. Note that communications and local computations may be performed in parallel and asynchronously.

## 3  Distributed optimization under global regularity

The most standard approach for distributing a first-order optimization method consists in computing a subgradient of the average function

$$\nabla \bar{f}(\theta) = \frac{1}{n} \sum_{i=1}^{n} \nabla f_i(\theta), \tag{8}$$

where $\nabla f_i(\theta)$ is any subgradient of $f_i$ at $\theta$, by sending the current parameter $\theta_t$ to all nodes, performing the computation of all local subgradients in parallel and averaging them on a master node. Since each iteration requires communicating twice to the whole network (once for $\theta_t$ and once for sending the local subgradients to the master node, which both take a time $\Delta\tau$ where $\Delta$ is the diameter of the network) and one subgradient computation (on each node and performed in parallel), the time to reach a precision $\varepsilon$ with such a distributed subgradient descent is upper-bounded by

$$O\left(\left(\frac{RL_g}{\varepsilon}\right)^2 (\Delta\tau + 1)\right). \tag{9}$$

Note that this convergence rate depends on the global Lipschitz constant $L_g$, and is thus applicable under the global regularity assumption. The number of subgradient computations in Eq. (9) (i.e. the term not proportional to $\tau$) cannot be improved, since it is already optimal for objective functions defined on only one machine (see for example Theorem 3.13 p. 280 in [18]). However, quite surprisingly, the error due to communication time may benefit from fast convergence rates in $O(RL_g/\varepsilon)$. This result is already known under the local regularity assumption (i.e. replacing $L_g$ with $L_\ell$ or even $\max_i L_i$) in the case of decentralized optimization [14] or distributed optimization on a totally connected network [17]. To our knowledge, the case of global regularity has not been investigated by prior work.

### 3.1  A simple algorithm with fast communication rates

We now show that the simple smoothing approach introduced in [10] can lead to fast rates for error due to communication time. Let $\gamma > 0$ and $f : \mathbb{R}^d \to \mathbb{R}$ be a real function. We denote as *smoothed version of $f$* the following function:

$$f^\gamma(\theta) = \mathbb{E}\left[f(\theta + \gamma X)\right], \tag{10}$$

where $X \sim \mathcal{N}(0, I)$ is a standard Gaussian random variable. The following lemma shows that $f^\gamma$ is both smooth and a good approximation of $f$.

**Algorithm 1** distributed randomized smoothing

---

**Input:** approximation error $\varepsilon > 0$, communication graph $\mathcal{G}$, $\alpha_0 = 1$, $\alpha_{t+1} = 2/(1 + \sqrt{1 + 4/\alpha_t^2})$
$T = \left\lceil \frac{20RL_g d^{1/4}}{\varepsilon} \right\rceil$, $K = \left\lceil \frac{5RL_g d^{-1/4}}{\varepsilon} \right\rceil$, $\gamma_t = Rd^{-1/4}\alpha_t$, $\eta_t = \frac{R\alpha_t}{2L_g(d^{1/4} + \sqrt{\frac{t+1}{K}})}$ .

**Output:** optimizer $\theta_T$
1: Compute a spanning tree $\mathcal{T}$ on $\mathcal{G}$.
2: Send a random seed $s$ to every node in $\mathcal{T}$.
3: Initialize the random number generator of each node using $s$.
4: $x_0 = 0$, $z_0 = 0$, $G_0 = 0$
5: **for** $t = 0$ to $T - 1$ **do**
6:     $y_t = (1 - \alpha_t)x_t + \alpha_t z_t$
7:     Send $y_t$ to every node in $\mathcal{T}$.
8:     Each node $i$ computes $g_i = \frac{1}{K}\sum_{k=1}^K \nabla f_i(y_t + \gamma_t X_{t,k})$, where $X_{t,k} \sim \mathcal{N}(0, I)$
9:     $G_{t+1} = G_t + \frac{1}{n\alpha_t}\sum_i g_i$
10:     $z_{t+1} = \text{argmin}_{x \in \mathcal{K}} \|x + \eta_{t+1}G_{t+1}\|_2^2$
11:     $x_{t+1} = (1 - \alpha_t)x_t + \alpha_t z_{t+1}$
12: **end for**
13: **return** $\theta_T = x_T$

---

**Lemma 1** (Lemma $E.3$ of [10]). *If $\gamma > 0$, then $f^\gamma$ is $\frac{L_g}{\gamma}$-smooth and, for all $\theta \in \mathbb{R}^d$,*

$$f(\theta) \leq f^\gamma(\theta) \leq f(\theta) + \gamma L_g \sqrt{d}. \tag{11}$$

Hence, smoothing the objective function allows the use of accelerated optimization algorithms and provides faster convergence rates. Of course, the price to pay is that each computation of the smoothed gradient $\nabla \bar{f}^\gamma(\theta) = \frac{1}{n}\sum_{i=1}^n \nabla f_i^\gamma(\theta)$ now requires, at each iteration $m$, to sample a sufficient amount of subgradients $\nabla f_i(\theta + \gamma X_{m,k})$ to approximate Eq. (10), where $X_{m,k}$ are $K$ i.i.d. Gaussian random variables. At first glance, this algorithm requires all computing units to synchronize on the choice of $X_{m,k}$, which would require to send to all nodes each $X_{m,k}$ and thus incur a communication cost proportional to the number of samples. Fortunately, computing units only need to share one random seed $s \in \mathbb{R}$ and then use a random number generator initialized with the provided seed to generate the same random variables $X_{m,k}$ without the need to communicate any vector. The overall algorithm, denoted *distributed randomized smoothing* (DRS), uses the randomized smoothing optimization algorithm of [10] adapted to a distributed setting, and is summarized in Alg. 1. The computation of a spanning tree $\mathcal{T}$ in step 1 allows efficient communication to the whole network in time at most $\Delta \tau$. Most of the algorithm (i.e. steps $2, 4, 6, 7, 9, 10$ and $11$) are performed on the root of the spanning subtree $\mathcal{T}$, while the rest of the computing units are responsible for computing the smoothed gradient (step 8). The seed $s$ of step 2 is used to ensure that every $X_{m,k}$, although random, is the *same on every node*. Finally, step 10 is a simple orthogonal projection of the gradient step on the convex set $\mathcal{K}$. We now show that the DRS algorithm converges to the optimal parameter under the global regularity assumption.

**Theorem 1.** *Under global regularity (A1), Alg. 1 achieves an approximation error $\mathbb{E}\left[\bar{f}(\theta_T)\right] - \bar{f}(\theta^*)$ of at most $\varepsilon > 0$ in a time $T_\varepsilon$ upper-bounded by*

$$O\left(\frac{RL_g}{\varepsilon}(\Delta\tau + 1)d^{1/4} + \left(\frac{RL_g}{\varepsilon}\right)^2\right). \tag{12}$$

More specifically, Alg. 1 completes its $T$ iterations by time

$$T_\varepsilon \leq 40\left\lceil \frac{RL_g d^{1/4}}{\varepsilon}\right\rceil \Delta\tau + 100\left\lceil \frac{RL_g d^{1/4}}{\varepsilon}\right\rceil \left\lceil \frac{RL_g d^{-1/4}}{\varepsilon}\right\rceil . \tag{13}$$

Comparing Eq. (13) to Eq. (9), we can see that our algorithm improves on the standard method when the dimension is not too large, and more specifically

$$d \leq \left(\frac{RL_g}{\varepsilon}\right)^4. \tag{14}$$

In practice, this condition is easily met, as $\varepsilon \leq 10^{-2}$ already leads to the condition $d \leq 10^8$ (assuming that $R$ and $L_g$ have values around 1). Moreover, for problems of moderate dimension, the term

$d^{1/4}$ remains a small multiplicative factor (e.g. for $d = 1000$, $d^{1/4} \approx 6$). Finally, note that DRS achieves a linear speedup when communication through the whole network requires a constant time, i.e., $\Delta\tau = O(1)$, and the convexity of each local function $f_i$ is not necessary for Theorem 1 to hold.

**Remark 1.** Several other smoothing methods exist in the literature, notably the *Moreau envelope* [19] enjoying a dimension-free approximation guarantee. However, the Moreau envelope of an average of functions is difficult to compute (requires a different oracle than computing a subgradient), and unfortunately leads to convergence rates with respect to local Lipschitz characteristics instead of $L_g$.

### 3.2 Optimal convergence rate

The following result provides oracle complexity lower bounds under the global regularity assumption, and is proved in the supplemental material. This lower bound extends the communication complexity lower bound for totally connected communication networks from [17].

**Theorem 2.** *Let $\mathcal{G}$ be a network of computing units of size $n > 0$, and $L_g, R > 0$. There exists $n$ functions $f_i : \mathbb{R}^d \to \mathbb{R}$ such that (A1) holds and, for any $t < \frac{(d-2)}{2}\min\{\Delta\tau, 1\}$ and any black-box procedure one has, for all $i \in \{1, ..., n\}$,*

$$\bar{f}(\theta_{i,t}) - \min_{\theta \in B_2(R)} \bar{f}(\theta) \geq \frac{RL_g}{36}\sqrt{\frac{1}{(1 + \frac{t}{2\Delta\tau})^2} + \frac{1}{1 + t}}. \tag{15}$$

Assuming that the dimension $d$ is large compared to the characteristic values of the problem (a standard set-up for lower bounds in non-smooth optimization [20, Theorem 3.2.1]), Theorem 2 implies that, under the global regularity assumption (A1), the time to reach a precision $\varepsilon > 0$ with any black-box procedure is lower bounded by

$$\Omega\left(\frac{RL_g}{\varepsilon}\Delta\tau + \left(\frac{RL_g}{\varepsilon}\right)^2\right), \tag{16}$$

where the notation $g(\varepsilon) = \Omega(f(\varepsilon))$ stands for $\exists C > 0$ s.t. $\forall \varepsilon > 0, g(\varepsilon) \geq Cf(\varepsilon)$. This lower bound proves that the convergence rate of DRS in Eq. (13) is optimal with respect to computation time and within a $d^{1/4}$ multiplicative factor of the optimal convergence rate with respect to communication.

The proof of Theorem 2 relies on the use of two objective functions: first, the standard worst case function used for single machine convex optimization (see e.g. [18]) is used to obtain a lower bound on the local computation time of individual machines. Then, a second function first introduced in [17] is split on the two most distant machines to obtain worst case communication times. By aggregating these two functions, a third one is obtained with the desired lower bound on the convergence rate. The complete proof is available as supplementary material. Finally, note that, due to its random nature, Alg. 1 is not *per se* a black-box procedure, and Theorem 2 does not apply to it. Lower bounds for random algorithms are more challenging and left for future work.

**Remark 2.** The lower bound also holds for the average of local parameters $\frac{1}{n}\sum_{i=1}^n \theta_i$, and more generally any parameter that can be computed using any vector of the local memories at time $t$: in Theorem 2, $\theta_{i,t}$ may be replaced by any $\theta_t$ such that

$$\theta_t \in \text{Span}\left(\bigcup_{i \in \mathcal{V}} \mathcal{M}_{i,t}\right). \tag{17}$$

## 4 Decentralized optimization under local regularity

In many practical scenarios, the network may be unknown or changing through time, and a local communication scheme is preferable to the *master/slave* approach of Alg. 1. Decentralized algorithms tackle this problem by replacing targeted communication by *local averaging* of the values of neighboring nodes [9]. More specifically, we now consider that, during a communication step, each machine $i$ broadcasts a vector $x_i \in \mathbb{R}^d$ to its neighbors, then performs a weighted average of the values received from its neighbors:

$$\text{node } i \text{ sends } x_i \text{ to his neighbors and receives } \sum_j W_{ji}x_j. \tag{18}$$

In order to ensure the efficiency of this communication scheme, we impose standard assumptions on the matrix $W \in \mathbb{R}^{n \times n}$, called the *gossip* matrix [9, 8]:

1. $W$ is symmetric and positive semi-definite,

2. The kernel of $W$ is the set of constant vectors: $\text{Ker}(W) = \text{Span}(\mathbb{1})$, where $\mathbb{1} = (1, ..., 1)^\top$,

3. $W$ is defined on the edges of the network: $W_{ij} \neq 0$ only if $i = j$ or $(i, j) \in \mathcal{E}$.

Note that these assumptions are implied by symmetry, stochasticity and positive eigengap on $I - W$.

## 4.1 Optimal convergence rate

Similarly to the smooth and strongly-convex case of [8], the lower bound on the optimal convergence rate is obtained by replacing the diameter of the network with $1/\sqrt{\gamma(W)}$, where $\gamma(W) = \lambda_{n-1}(W)/\lambda_1(W)$ is the ratio between smallest non-zero and largest eigenvalues of $W$, also known as the *normalized eigengap*.

**Theorem 3.** *Let $L_\ell, R > 0$ and $\gamma \in (0, 1]$. There exists a matrix $W$ of eigengap $\gamma(W) = \gamma$, and $n$ functions $f_i$ satisfying (A2), where $n$ is the size of $W$, such that for all $t < \frac{d-2}{2} \min(\tau/\sqrt{\gamma}, 1)$ and all $i \in \{1, ..., n\}$,*

$$\bar{f}(\theta_{i,t}) - \min_{\theta \in B_2(R)} \bar{f}(\theta) \geq \frac{RL_\ell}{108} \sqrt{\frac{1}{(1 + \frac{2t\sqrt{\gamma}}{\tau})^2} + \frac{1}{1+t}}. \tag{19}$$

Assuming that the dimension $d$ is large compared to the characteristic values of the problem, Theorem 3 implies that, under the local regularity assumption (A2) and for a gossip matrix $W$ with eigengap $\gamma(W)$, the time to reach a precision $\varepsilon > 0$ with any *decentralized* black-box procedure is lower-bounded by

$$\Omega\left(\frac{RL_\ell}{\varepsilon} \frac{\tau}{\sqrt{\gamma(W)}} + \left(\frac{RL_\ell}{\varepsilon}\right)^2\right). \tag{20}$$

The proof of Theorem 3 relies on linear graphs (whose diameter is proportional to $1/\sqrt{\gamma(L)}$ where $L$ is the Laplacian matrix) and Theorem 2. More specifically, a technical aspect of the proof consists in splitting the functions used in Theorem 2 on multiple nodes to obtain a dependency in $L_\ell$ instead of $L_g$. The complete derivation is available as supplementary material.

## 4.2 Optimal decentralized algorithm

We now provide an optimal decentralized optimization algorithm under (A2). This algorithm is closely related to the primal-dual algorithm proposed by [14], which we modify by the use of accelerated gossip using Chebyshev polynomials as in [8].

First, we formulate our optimization problem in Eq. (1) as the saddle-point problem in Eq. (21) below, by considering the equivalent problem of minimizing $\frac{1}{n} \sum_{i=1}^{n} f_i(\theta_i)$ over $\Theta = (\theta_1, \ldots, \theta_n) \in \mathcal{K}^n$ with the constraint that $\theta_1 = \cdots = \theta_n$, or equivalently $\Theta A = 0$, where $A$ is a square root of the symmetric matrix $W$. Through Lagrangian duality, we therefore get the equivalent problem:

$$\min_{\Theta \in \mathcal{K}^n} \max_{\Lambda \in \mathbb{R}^{d \times n}} \frac{1}{n} \sum_{i=1}^{n} f_i(\theta_i) - \text{tr}\, \Lambda^\top \Theta A. \tag{21}$$

We solve it by applying Algorithm 1 in Chambolle-Pock [21] (we could alternatively apply composite Mirror-Prox [22]), which is both simple and well tailored to our problem: (a) it is an accelerated method for saddle-point problems, (b) it allows for composite problems with a sum of non-smooth and smooth terms, (c) it provides a primal-dual gap that can easily be extended to the case of approximate proximal operators. At each iteration $t$, with initialization $\Lambda^0 = 0$ and $\Theta^0 = \Theta^{-1} = (\theta_0, \ldots, \theta_0)$:

$$
\begin{aligned}
(a) \quad & \Lambda^{t+1} = \Lambda^t - \sigma(2\Theta^{t+1} - \Theta^t)A \\
(b) \quad & \Theta^{t+1} = \underset{\Theta \in \mathcal{K}^n}{\text{argmin}} \frac{1}{n} \sum_{i=1}^{n} f_i(\theta_i) - \text{tr}\, \Theta^\top \Lambda^{t+1} A^\top + \frac{1}{2\eta} \text{tr}(\Theta - \Theta^t)^\top(\Theta - \Theta^t),
\end{aligned} \tag{22}
$$

where the gain parameters $\eta, \sigma$ are required to satisfy $\sigma\eta\lambda_1(W) \leq 1$. We implement the algorithm with the variables $\Theta^t$ and $Y^t = \Lambda^t A^\top = (y_1^t, \ldots, y_n^t) \in \mathbb{R}^{d \times n}$, for which all updates can be made

---

**Algorithm 2** multi-step primal-dual algorithm

---

**Input:** approximation error $\varepsilon > 0$, gossip matrix $W \in \mathbb{R}^{n \times n}$,

$$K = \lfloor 1/\sqrt{\gamma(W)} \rfloor, M = T = \lceil \frac{4RL_\ell}{\varepsilon} \rceil, c_1 = \frac{1-\sqrt{\gamma(W)}}{1+\sqrt{\gamma(W)}}, \eta = \frac{nR}{L_\ell}\frac{1-c_1^K}{1+c_1^K}, \sigma = \frac{1+c_1^{2K}}{\tau(1-c_1^K)^2}.$$

**Output:** optimizer $\bar{\theta}_T$

1: $\Theta_0 = 0, \Theta_{-1} = 0, Y_0 = 0$
2: **for** $t = 0$ to $T - 1$ **do**
3:     $Y^{t+1} = Y^t - \sigma \text{ ACCELERATEDGOSSIP}(2\Theta^t - \Theta^{t-1}, W, K)$          // see [8, Alg. 2]
4:     $\tilde{\Theta}^0 = \Theta^t$
5:     **for** $m = 0$ to $M - 1$ **do**
6:         $\tilde{\theta}_i^{m+1} = \frac{m}{m+2}\tilde{\theta}_i^m - \frac{2}{m+2}\left[\frac{\eta}{n}\nabla f_i(\tilde{\theta}_i^m) - \eta y_i^{t+1} - \theta_i^t\right], \forall i \in \{1, \ldots, n\}$
7:     **end for**
8:     $\Theta^{t+1} = \tilde{\Theta}^M$
9: **end for**
10: **return** $\bar{\theta}_T = \frac{1}{T}\frac{1}{n}\sum_{t=1}^T \sum_{i=1}^n \theta_i^t$

---

locally: Since $AA^\top = W$, they now become

$$
\begin{aligned}
(a') \quad Y^{t+1} &= Y^t - \sigma(2\Theta^{t+1} - \Theta^t)W \\
(b') \quad \theta_i^{t+1} &= \underset{\theta_i \in \mathcal{K}}{\text{argmin}} \frac{1}{n}f_i(\theta_i) - \theta_i^\top y_i^{t+1} + \frac{1}{2\eta}\|\theta_i - \theta_i^t\|^2, \forall i \in \{1, \ldots, n\},
\end{aligned}
\tag{23}
$$

The step $(b')$ still requires a proximal step for each function $f_i$. We approximate it by the outcome of the subgradient method run for $M$ steps, with a step-size proportional to $2/(m+2)$ as suggested in [23]. That is, initialized with $\tilde{\theta}_i^0 = \theta_i^t$, it performs the iterations

$$\tilde{\theta}_i^{m+1} = \frac{m}{m+2}\tilde{\theta}_i^m - \frac{2}{m+2}\left[\frac{\eta}{n}\nabla f_i(\tilde{\theta}_i^m) - \eta y_i^{t+1} - \theta_i^t\right], \quad m = 0, \ldots, M - 1. \tag{24}$$

We thus replace the step $(b')$ by running $M$ steps of the subgradient method to obtain $\tilde{\theta}_i^M$.

**Theorem 4.** *Under local regularity (A2), the approximation error with the iterative algorithm of Eq. (23) and (24) after $T$ iterations and using $M$ subgradient steps per iteration is bounded by*

$$\bar{f}(\bar{\theta}_T) - \min_{\theta \in \mathcal{K}} \bar{f}(\theta) \leq \frac{RL_\ell}{\sqrt{\gamma(W)}}\left(\frac{1}{T} + \frac{1}{M}\right). \tag{25}$$

Theorem 4 implies that the proposed algorithm achieves an error of at most $\varepsilon$ in a time no larger than

$$O\left(\frac{RL_\ell}{\varepsilon}\frac{\tau}{\sqrt{\gamma(W)}} + \left(\frac{RL_\ell}{\varepsilon}\frac{1}{\sqrt{\gamma(W)}}\right)^2\right). \tag{26}$$

While the first term (associated to communication) is optimal, the second does not match the lower bound of Theorem 3. This situation is similar to that of strongly-convex and smooth decentralized optimization [8], when the number of communication steps is taken equal to the number of overall iterations.

By using Chebyshev acceleration [24, 25] with an increased number of communication steps, the algorithm reaches the optimal convergence rate. More precisely, since one communication step is a multiplication (of $\Theta$ e.g.) by the gossip matrix $W$, performing $K$ communication steps is equivalent to multiplying by a power of $W$. More generally, multiplication by any polynomial $P_K(W)$ of degree $K$ can be achieved in $K$ steps. Since our algorithm depends on the eigengap of the gossip matrix, a good choice of polynomial consists in maximizing this eigengap $\gamma(P_K(W))$. This is the approach followed by [8] and leads to the choice $P_K(x) = 1 - T_K(c_2(1-x))/T_K(c_2)$, where $c_2 = (1 + \gamma(W))/(1 - \gamma(W))$ and $T_K$ are the Chebyshev polynomials [24] defined as $T_0(x) = 1$, $T_1(x) = x$, and, for all $k \geq 1$, $T_{k+1}(x) = 2xT_k(x) - T_{k-1}(x)$. We refer the reader to [8] for more details on the method. Finally, as mentioned in [8], chosing $K = \lfloor 1/\sqrt{\gamma(W)} \rfloor$ leads to an eigengap $\gamma(P_K(W)) \geq 1/4$ and the optimal convergence rate.

We denote the resulting algorithm as *multi-step primal-dual* (MSPD) and describe it in Alg. 2. The procedure ACCELERATEDGOSSIP is extracted from [8, Algorithm 2] and performs one step of Chebyshev accelerated gossip, while steps $4$ to $8$ compute the approximation of the minimization problem (b') of Eq. (23). Our performance guarantee for the MSPD algorithm is then the following:

**Theorem 5.** *Under local regularity (A2), Alg. 2 achieves an approximation error $\bar{f}(\bar{\theta}_T) - \bar{f}(\theta^*)$ of at most $\varepsilon > 0$ in a time $T_\varepsilon$ upper-bounded by*

$$O\left(\frac{RL_\ell}{\varepsilon}\frac{\tau}{\sqrt{\gamma(W)}} + \left(\frac{RL_\ell}{\varepsilon}\right)^2\right), \tag{27}$$

*which matches the lower complexity bound of Theorem 3. Alg. 2 is therefore optimal under the the local regularity assumption (A2).*

**Remark 3.** It is clear from the algorithm's description that it completes its $T$ iterations by time

$$T_\varepsilon \leq \left\lceil\frac{4RL_\ell}{\varepsilon}\right\rceil\frac{\tau}{\sqrt{\gamma(W)}} + \left\lceil\frac{4RL_\ell}{\varepsilon}\right\rceil^2. \tag{28}$$

To obtain the average of local parameters $\bar{\theta}_T = \frac{1}{nT}\sum_{t=1}^T\sum_{i=1}^n\theta_i$, one can then rely on the gossip algorithm [9] to average over the network the individual nodes' time averages. Let $W' = I - c_3P_K(W)$ where $c_3 = (1 + c_1^{2K})/(1 - c_1^K)^2$. Since $W'$ is bi-stochastic, semi-definite positive and $\lambda_2(W') = 1 - \gamma(P_K(W)) \leq 3/4$, using it for gossiping the time averages leads to a time $O\left(\frac{\tau}{\sqrt{\gamma}}\ln\left(\frac{RL_\ell}{\varepsilon}\right)\right)$ to ensure that each node reaches a precision $\varepsilon$ on the objective function (see [9] for more details on the linear convergence of gossip), which is negligible compared to Eq. (27).

**Remark 4.** A stochastic version of the algorithm is also possible by considering stochastic oracles on each $f_i$ and using stochastic subgradient descent instead of the subgradient method.

**Remark 5.** In the more general context where node compute times $\rho_i$ are not necessarily all equal to 1, we may still apply Alg. 2, where now the number of subgradient iterations performed by node $i$ is $M/\rho_i$ rather than $M$. The proof of Theorem 5 also applies, and now yields the modified upper bound on time to reach precision $\varepsilon$:

$$O\left(\frac{RL_\ell}{\varepsilon}\frac{\tau}{\sqrt{\gamma(W)}} + \left(\frac{RL_c}{\varepsilon}\right)^2\right), \tag{29}$$

where $L_c^2 = \frac{1}{n}\sum_{i=1}^n\rho_iL_i^2$.

## 5   Conclusion

In this paper, we provide optimal convergence rates for non-smooth and convex distributed optimization in two settings: Lipschitz continuity of the *global* objective function, and Lipschitz continuity of *local* individual functions. Under the *local* regularity assumption, we provide optimal convergence rates that depend on the $\ell_2$-average of the local Lipschitz constants and the (normalized) eigengap of the gossip matrix. Moreover, we also provide the first optimal decentralized algorithm, called *multi-step primal-dual* (MSPD).

Under the *global* regularity assumption, we provide a lower complexity bound that depends on the Lipschitz constant of the (global) objective function, as well as a distributed version of the smoothing approach of [10] and show that this algorithm is within a $d^{1/4}$ multiplicative factor of the optimal convergence rate.

In both settings, the optimal convergence rate exhibits two different speeds: a slow rate in $\Theta(1/\sqrt{t})$ with respect to local computations and a fast rate in $\Theta(1/t)$ due to communication. Intuitively, communication is the limiting factor in the initial phase of optimization. However, its impact decreases with time and, for the final phase of optimization, local computation time is the main limiting factor.

The analysis presented in this paper allows several natural extensions, including time-varying communication networks, asynchronous algorithms, stochastic settings, and an analysis of unequal node compute speeds going beyond Remark 5. Moreover, despite the efficiency of DRS, finding an optimal algorithm under the global regularity assumption remains an open problem and would make a notable addition to this work.

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
