[Supplementary Material]

# Optimal Algorithms for Non-Smooth Distributed Optimization in Networks

# SUPPLEMENTARY MATERIAL

**Kevin Scaman**[1]  **Francis Bach**[2]  **Sébastien Bubeck**[3]  **Yin Tat Lee**[3,4]  **Laurent Massoulié**[2,5]
[1] Huawei Noah's Ark Lab, [2] INRIA, Ecole Normale Supérieure, PSL Research University,
[3] Microsoft Research, [4] University of Washington, [5] MSR-INRIA Joint Centre

## Abstract

This supplementary document contains complete proofs of the theorems presented in the article "Optimal Algorithms for Non-Smooth Distributed Optimization in Networks".

## 1   Proof of the convergence rate of DRS (Theorem 1)

Corollary 2.4 of [1] gives, with the appropriate choice of gradient step $\eta_t$ and smoothing $\gamma_t$,

$$\mathbb{E}\left[\bar{f}(\theta_T)\right] - \min_{\theta \in \mathcal{K}} \bar{f}(\theta) \leq \frac{10 R L_g d^{1/4}}{T} + \frac{5 R L_g}{\sqrt{TK}}. \tag{1}$$

Thus, to reach a precision $\varepsilon > 0$, we may set $T = \left\lceil \frac{20 R L_g d^{1/4}}{\varepsilon} \right\rceil$ and $K = \left\lceil \frac{5 R L_g d^{-1/4}}{\varepsilon} \right\rceil$, leading to the desired bound on the time $T_\varepsilon = T(2\Delta\tau + K)$ to reach $\varepsilon$.

## 2   Proof of the lower bound under global regularity (Theorem 2)

Let $i_0 \in \mathcal{V}$ and $i_1 \in \mathcal{V}$ be two nodes at distance $\Delta$. The function used by [2] to prove the oracle complexity for Lipschitz and bounded functions is

$$g_1(\theta) = \delta \max_{i \in \{1,...,t\}} \theta_i + \frac{\alpha}{2} \|\theta\|_2^2. \tag{2}$$

By considering this function on a single node (e.g. $i_0$), at least $O\left(\left(\frac{RL}{\varepsilon}\right)^2\right)$ subgradients will be necessary to obtain a precision $\varepsilon$. Moreover, we also split the difficult function used in [3]

$$g_2(\theta) = \gamma \sum_{i=1}^{t} |\theta_{i+1} - \theta_i| - \beta\theta_1 + \frac{\alpha}{2}\|\theta\|_2^2, \tag{3}$$

on the two extremal nodes $i_0$ and $i_1$ in order to ensure that communication is necessary between the most distant nodes of the network. The final function that we consider is, for all $i \in \{1,...,n\}$,

$$f_i(\theta) = \begin{cases} \gamma \sum_{i=1}^{k} |\theta_{2i} - \theta_{2i-1}| + \delta \max_{i \in \{2k+2,...,2k+1+l\}} \theta_i & \text{if } i = i_0 \\ \gamma \sum_{i=1}^{k} |\theta_{2i+1} - \theta_{2i}| - \beta\theta_1 + \frac{\alpha}{2}\|\theta\|_2^2 & \text{if } i = i_1 \\ 0 & \text{otherwise} \end{cases}, \tag{4}$$

where $\gamma, \delta, \beta, \alpha > 0$ and $k, l \geq 0$ are parameters of the function satisfying $2k + l < d$. The objective function is thus

$$\bar{f}(\theta) = \frac{1}{n} \left[ \gamma \sum_{i=1}^{2k} |\theta_{i+1} - \theta_i| - \beta\theta_1 + \delta \max_{i \in \{2k+2,\ldots,2k+1+l\}} \theta_i + \frac{\alpha}{2} \|\theta\|_2^2 \right] \qquad (5)$$

First, note that reordering the coordinates of $\theta$ between $\theta_2$ and $\theta_{2k+1}$ in a decreasing order can only decrease the value function $\bar{f}(\theta)$. Hence, the optimal value $\theta^*$ verifies this constraint and

$$\bar{f}(\theta^*) = \frac{1}{n} \left[ -\gamma\theta_{2k+1}^* - (\beta - \gamma)\theta_1^* + \delta \max_{i \in \{2k+2,\ldots,2k+1+l\}} \theta_i^* + \frac{\alpha}{2} \|\theta^*\|_2^2 \right]. \qquad (6)$$

Moreover, at the optimum, all the coordinates between $\theta_2$ and $\theta_{2k+1}$ are equal, all the coordinates between $\theta_{2k+2}$ and $\theta_{2k+1+l}$ are also equal, and all the coordinates after $\theta_{2k+1+l}$ are zero. Hence

$$\bar{f}(\theta^*) = \frac{1}{n} \left[ -\gamma\theta_{2k+1}^* - (\beta - \gamma)\theta_1^* + \delta\theta_{2k+2}^* + \frac{\alpha}{2} \left( {\theta_1^*}^2 + 2k{\theta_{2k+1}^*}^2 + l{\theta_{2k+2}^*}^2 \right) \right], \qquad (7)$$

and optimizing over $\theta_1^* \geq \theta_{2k+1}^* \geq 0 \geq \theta_{2k+2}^*$ leads to, when $\beta \geq \gamma(1 + \frac{1}{2k})$,

$$\bar{f}(\theta^*) = \frac{-1}{2\alpha n} \left[ (\beta - \gamma)^2 + \frac{\gamma^2}{2k} + \frac{\delta^2}{l} \right]. \qquad (8)$$

Now note that, starting from $\theta_0 = 0$, each subgradient step can only increase the number of non-zero coordinates between $\theta_{2k+2}$ and $\theta_{2k+1+l}$ by at most one. Thus, when $t < l$, we have

$$\max_{i \in \{2k+2,\ldots,2k+1+l\}} \theta_{t,i} \geq 0 . \qquad (9)$$

Moreover, increasing the number of non-zero coordinates between $\theta_1$ and $\theta_{2k+1}$ requires at least one subgradient step and $\Delta$ communication steps. As a result, when $t < \min\{l, 2k\Delta\tau\}$, we have $\theta_{t,2k+1} = 0$ and

$$\begin{aligned}\bar{f}(\theta_t) &\geq \min_{\theta \in \mathbb{R}^d} \frac{1}{n} \left[ -(\beta - \gamma)\theta_1 + \frac{\alpha}{2} \|\theta\|_2^2 \right] \\ &\geq \frac{-(\beta - \gamma)^2}{2\alpha n} .\end{aligned} \qquad (10)$$

Hence, we have, for $t < \min\{l, 2k\Delta\tau\}$,

$$\bar{f}(\theta_t) - \bar{f}(\theta^*) \geq \frac{1}{2\alpha n} \left[ \frac{\gamma^2}{2k} + \frac{\delta^2}{l} \right]. \qquad (11)$$

Optimizing $\bar{f}$ over a ball of radius $R \geq \|\theta^*\|_2$ thus leads to the previous approximation error bound, and we choose

$$R = \|\theta^*\|_2 = \frac{1}{\alpha^2} \left[ (\beta - \gamma)^2 + \frac{\gamma^2}{2k} + \frac{\delta^2}{l} \right]. \qquad (12)$$

Finally, the Lipschitz constant of the objective function $\bar{f}$ is

$$L_g = \frac{1}{n} \left[ \beta + 2\sqrt{2k+1}\gamma + \delta + \alpha R \right], \qquad (13)$$

and setting the parameters of $\bar{f}$ to $\beta = \gamma(1 + \frac{1}{\sqrt{2k}})$, $\delta = \frac{L_g n}{9}$, $\gamma = \frac{L_g n}{9\sqrt{k}}$, $l = \lfloor t \rfloor + 1$, and $k = \lfloor \frac{t}{2\Delta\tau} \rfloor + 1$ leads to $t < \min\{l, 2k\Delta\tau\}$ and

$$\bar{f}(\theta_t) - \bar{f}(\theta^*) \geq \frac{RL_g}{36} \sqrt{\frac{1}{(1 + \frac{t}{2\Delta\tau})^2} + \frac{1}{1 + t}}, \qquad (14)$$

while $\bar{f}$ is $L$-Lipschitz and $\|\theta^*\|_2 \leq R$.

## 3 Proof of the lower bound under local regularity (Theorem 3)

Following the idea introduced in [4], we prove Theorem 3 by applying Theorem 2 on linear graphs and splitting the local functions of Eq. (4) on multiple nodes to obtain $L_g \approx L_\ell$.

**Lemma 1.** *Let $\gamma \in (0,1]$. There exists a graph $\mathcal{G}_\gamma$ of size $n_\gamma$ and a gossip matrix $W_\gamma \in \mathbb{R}^{n_\gamma \times n_\gamma}$ on this graph such that $\gamma(W_\gamma) = \gamma$ and*

$$\gamma \geq \frac{2}{(n_\gamma + 1)^2}. \tag{15}$$

*When $\gamma \geq 1/3$, $\mathcal{G}_\gamma$ is a totally connected graph of size $n_\gamma = 3$. Otherwise, $\mathcal{G}_\gamma$ is a linear graph of size $n_\gamma \geq 3$.*

*Proof.* First of all, when $\gamma \geq 1/3$, we consider the totally connected network of 3 nodes, reweight only the edge $(v_1, v_3)$ by $a \in [0,1]$, and let $W_a$ be its Laplacian matrix. If $a = 1$, then the network is totally connected and $\gamma(W_a) = 1$. If, on the contrary, $a = 0$, then the network is a linear graph and $\gamma(W_a) = 1/3$. Thus, by continuity of the eigenvalues of a matrix, there exists a value $a \in [0,1]$ such that $\gamma(W_a) = \gamma$ and Eq. (15) is trivially verified. Otherwise, let $x_n = \frac{1-\cos(\frac{\pi}{n})}{1+\cos(\frac{\pi}{n})}$ be a decreasing sequence of positive numbers. Since $x_3 = 1/3$ and $\lim_n x_n = 0$, there exists $n_\gamma \geq 3$ such that $x_{n_\gamma} \geq \gamma > x_{n_\gamma+1}$. Let $\mathcal{G}_\gamma$ be the linear graph of size $n_\gamma$ ordered from node $v_1$ to $v_{n_\gamma}$, and weighted with $w_{i,i+1} = 1 - a\mathbb{1}\{i = 1\}$. If we take $W_a$ as the Laplacian of the weighted graph $\mathcal{G}_\gamma$, a simple calculation gives that, if $a = 0$, $\gamma(W_a) = x_{n_\gamma}$ and, if $a = 1$, the network is disconnected and $\gamma(W_a) = 0$. Thus, there exists a value $a \in [0,1]$ such that $\gamma(W_a) = \gamma$. Finally, by definition of $n_\gamma$, one has $\gamma > x_{n_\gamma+1} \geq \frac{2}{(n_\gamma+1)^2}$. $\qquad\square$

Let $\gamma \in (0,1]$ and $\mathcal{G}_\gamma$ the graph of Lemma 1. We now consider $I_0 = \{1, ..., m\}$ and $I_1 = \{n_\gamma - m + 1, ..., n_\gamma\}$ where $m = \lfloor \frac{n_\gamma+1}{3} \rfloor$. When $\gamma < 1/3$, the distance $d(I_0, I_1)$ between the two sets $I_0$ and $I_1$ is thus bounded by

$$d(I_0, I_1) = n_\gamma - 2m + 1 \geq \frac{n_\gamma + 1}{3}, \tag{16}$$

and we have

$$\frac{1}{\sqrt{\gamma}} \leq \frac{3d(I_0, I_1)}{\sqrt{2}}. \tag{17}$$

Moreover, Eq. (17) also trivially holds when $\gamma \geq 1/3$. We now consider the local functions of Eq. (4) splitted on $I_0$ and $I_1$:

$$f_i(\theta) = \begin{cases} \frac{1}{m}\left[\gamma \sum_{i=1}^k |\theta_{2i} - \theta_{2i-1}| + \delta \max_{i \in \{2k+2,...,2k+1+l\}} \theta_i\right] & \text{if } i \in I_0 \\ \frac{1}{m}\left[\gamma \sum_{i=1}^k |\theta_{2i+1} - \theta_{2i}| - \beta\theta_1 + \frac{\alpha}{2}\|\theta\|_2^2\right] & \text{if } i \in I_1 \\ 0 & \text{otherwise} \end{cases} \tag{18}$$

The average function $\bar{f}$ remains unchanged and the time to communicate a vector between a node of $I_0$ and a node of $I_1$ is at least $d(I_0, I_1)\tau$. Thus, the same result as Theorem 2 holds with $\Delta = d(I_0, I_1)$. We thus have

$$\bar{f}(\theta_{i,t}) - \min_{\theta \in B_2(R)} \bar{f}(\theta) \geq \frac{RL_g}{36}\sqrt{\frac{1}{(1 + \frac{t}{2d(I_0,I_1)\tau})^2} + \frac{1}{1+t}}. \tag{19}$$

Finally, the local Lipschitz constant $L_\ell$ is bounded by

$$L_\ell \leq \sqrt{\frac{n_\gamma}{m}} L_g \leq 3L_g, \tag{20}$$

and Eq. (17), Eq. (19) and Eq. (20) lead to the desired result.

## 4 Proof of the convergence rate of MSPD (Theorem 4 and Theorem 5)

Theorem 1 (b) in [5] implies that, provided $\tau\sigma\lambda_1(W) < 1$, the algorithm with exact proximal step leads to a restricted primal-dual gap

$$\sup_{\|\Lambda'\|_F \leq c} \left\{ \frac{1}{n}\sum_{i=1}^n f_i(\theta_i) - \operatorname{tr}\Lambda'^\top \Theta A \right\} - \inf_{\Theta' \in \mathcal{K}^n} \left\{ \frac{1}{n}\sum_{i=1}^n f_i(\theta'_i) - \operatorname{tr}\Lambda^\top \Theta' A \right\}$$

of
$$\varepsilon = \frac{1}{2t}\left(\frac{nR^2}{\eta} + \frac{c^2}{\sigma}\right).$$
This implies that our candidate $\Theta$ is such that
$$\frac{1}{n}\sum_{i=1}^{n} f_i(\theta_i) + c\|\Theta A\|_F \le \inf_{\Theta' \in \mathcal{K}^n} \left\{\frac{1}{n}\sum_{i=1}^{n} f_i(\theta_i') + c\|\Theta' A\|_F + \varepsilon\right\}.$$

Let $\theta$ be the average of all $\theta_i$. We have:
$$\frac{1}{n}\sum_{i=1}^{n} f_i(\theta) \le \frac{1}{n}\sum_{i=1}^{n} f_i(\theta_i) + \frac{1}{n}\sum_{i=1}^{n} L_i\|\theta_i - \theta\| \le \frac{1}{n}\sum_{i=1}^{n} f_i(\theta_i) + \frac{1}{\sqrt{n}}\sqrt{\frac{1}{n}\sum_{i=1}^{n} L_i^2} \cdot \left\|\Theta(I - \mathbf{1}\mathbf{1}^\top/n)\right\|_F$$
$$\le \frac{1}{n}\sum_{i=1}^{n} f_i(\theta_i) + \frac{1}{\sqrt{n}}\sqrt{\frac{\frac{1}{n}\sum_{i=1}^{n} L_i^2}{\lambda_{n-1}(W)}} \cdot \left\|\Theta A\right\|_F.$$

Thus, if we take $c = \frac{1}{\sqrt{n}}\sqrt{\frac{\frac{1}{n}\sum_{i=1}^{n} L_i^2}{\lambda_{n-1}(W)}}$, we obtain
$$\frac{1}{n}\sum_{i=1}^{n} f_i(\theta) \le \frac{1}{n}\sum_{i=1}^{n} f_i(\theta_*) + \varepsilon,$$
and we thus obtain a $\varepsilon$-minimizer of the original problem.

We have
$$\varepsilon \le \frac{1}{2T}\left(\frac{nR^2}{\eta} + \frac{\frac{1}{\lambda_{n-1}(W)}\frac{1}{n^2}\sum_{i=1}^{n} L_i^2}{\sigma}\right)$$
with the constraint $\sigma\eta\lambda_1(W) < 1$. This leads to, with the choice
$$\eta = nR\sqrt{\frac{\lambda_{n-1}(W)/\lambda_1(W)}{\sum_{i=1}^{n} L_i^2/n}}$$
and taking $\sigma$ to the limit $\sigma\eta\lambda_1(W) = 1$, to a convergence rate of
$$\varepsilon = \frac{1}{T}R\sqrt{\frac{1}{n}\sum_{i=1}^{n} L_i^2}\sqrt{\frac{\lambda_1(W)}{\lambda_{n-1}(W)}}.$$

Since we cannot use the exact proximal operator of $f_i$, we instead approximate it. If we approximate (with the proper notion of gap [5, Eq. (11)]) each $\operatorname{argmin}_{\theta_i \in \mathcal{K}} f_i(\theta_i) + \frac{n}{2\eta}\|\theta_i - z\|^2$ up to $\delta_i$, then the overall added gap is $\frac{1}{n}\sum_{i=1}^{n} \delta_i$. If we do $M$ steps of subgradient steps then the associated gap is $\delta_i = \frac{L_i^2\eta}{nM}$ (standard result for strongly-convex subgradient [6]). Therefore the added gap is
$$\frac{1}{M}R\sqrt{\frac{1}{n}\sum_{i=1}^{n} L_i^2}\sqrt{\frac{\lambda_1(W)}{\lambda_{n-1}(W)}}.$$

Therefore after $T$ communication steps, i.e., communication time $T\tau$ plus $MT$ subgradient evaluations, i.e., time $MT$, we get an error of
$$\left(\frac{1}{T} + \frac{1}{M}\right)\frac{RL_\ell}{\sqrt{\gamma}},$$
where $\gamma = \gamma(W) = \lambda_{n-1}(W)/\lambda_1(W)$. Thus to reach $\varepsilon$, it takes
$$\left\lceil\frac{2RL_\ell}{\varepsilon}\frac{1}{\sqrt{\gamma}}\right\rceil\tau + \left\lceil\frac{4RL_\ell}{\varepsilon}\frac{1}{\sqrt{\gamma}}\right\rceil^2.$$

The second term is optimal, while the first term is not. We therefore do accelerated gossip instead of plain gossip. By performing $K$ steps of gossip instead of one, with $K = \lfloor 1/\sqrt{\gamma}\rfloor$, the eigengap is lower bounded by $\gamma(P_K(W)) \ge 1/4$, and the overall time to obtain an error below $\varepsilon$ becomes
$$\left\lceil\frac{4RL_\ell}{\varepsilon}\right\rceil\frac{\tau}{\sqrt{\gamma(W)}} + \left\lceil\frac{4RL_\ell}{\varepsilon}\right\rceil^2,$$
as announced.