[Reviews · NeurIPS 2018]

Reviewer 1



The paper studies distributed algorithms for non-smooth convex optimization. The authors consider a standard distributed optimization setting in which given a communication graph, where each node has access to its own non-smooth convex function, the goal is to globally minimize the average of functions in little as possible time, where time accounts for both computation on each node and communication between neighboring nodes in the graph. The authors consider two natural assumptions, 1) where the global function (i.e., average of local functions) is Lipschitz continuous and 2) when each local function is Lipschitz continuous. They also consider both distributed master/slave-type algorithms and completely decentralized gossip-based algorithms. In the centralized master/slave settings, the authors show that under the global Lipschitz assumption, a distributed version of Nesterov's accelerated gradient method (slave nodes only compute gradients and send to master node which runs Nesterov's method), coupled with a standard randomized smoothing technique, gives optimal runtime bounds up to an extra multiplicative dependence on d^{1/4} (d is the dimension), which is caused due to the use of the randomized smoothing. They prove the corresponding lower-bound. In the decentralized setting, the authors use a known primal-dual algorithm which is distributed via a gossip-based algorithm which overall gives optimal runtime (they prove a corresponding tight lower-bound). Overall, I think the paper presents very solid contributions on problems of definite interest to the optimization community in NIPS. Both upper bounds and nearly matching lower bounds are given, and hence I support its acceptance. On the technical side, it seems the results are heavily based on previous results, but still I think the current paper has clear contributions. One thing is, that I think the writing of Section 4 and in particular Subsection 4.2. could be greatly improved and I encourage the authors to work on the presentation. For one, the authors introduce the saddle-point formulation in Eq. 21 which accounts for the constraint that the solutions on all nodes should be equal. However, their algorithm always returns the average of the solutions on all nodes, so why should we enforce this equality constraint at the first-place? a follow-up question, is then why do we need to use the primal-dual method of [14]? I would be very happy to see more technical explanations here to the algorithmic choices/arguments made in the design of Algorithm 2. Another question regarding the master/slave algorithm: is it trivial to apply a similar technique when the non-smooth functions admit known DETERMINISTIC smoothing schemes (though not necessarily proximal-friendly)? *** post rebuttal comments *** I have read the response and it answered my questions.

Reviewer 2



The paper considers the question of optimizing the average of local non-smooth convex functions in a distributed network of computers. The study of distributed optimization algorithms is a relevant problem, and the importance of this work consists in proposing the first optimal algorithm for decentralized distributed non-smooth convex optimization under a local regularity assumption. To achieve this goal, the authors combine an accelerated gossip step and a primal-dual algorithm. Moreover, the authors propose an almost optimal algorithm under a weaker global regularity assumption, which is based in a randomized smoothing step. The authors establish lower and upper bounds in complexity of both problems, which match for the local regularity assumption. This paper is an important contribution as they are the first in establishing optimality in non-smooth distributed optimization, which is an important setting in modern applications. The result is achieved by combining existing tools in a clever manner. The paper is well written and easy to follow. I found the theoretical results well motivated and explained.

Reviewer 3



This paper investigates the distributed optimization of non-smooth convex function using a network of computing units. Both the global and the local Lipschitz continuities are considered. An optimal algorithm is proposed under the local Lipschitz continuity, while a simple but efficient algorithm is proposed for the global Lipschitz continuity. Both results are interesting to me. A few questions need to be clarified in rebuttal: - Eq. (9), is the upper bound of the time complexity to achieve precision epsilon. Is the *precision* for deterministic measure or expectation measure? I recall that it should be deterministic measure. If so, the comparison between your result (13) and existing result (9) may be not be fair enough. - In Theorem 1, it is not very clear how the number of working units affect the time complexity? It would be interesting to discuss if or when the linear speedup can be achieved. - Alg1 is essentially the accelerated dual averaging algorithm plus the smoothing trick. But I am not sure if using the Gaussian smoother is the best way, since it brings additional uncertainty to estimate the gradient of the smoothed function. What if using the uniform smoother like the one "A comprehensive linear speedup analysis for asynchronous stochastic parallel optimization from zeroth-order to first-order, NIPS, 2016", that can exactly obtain the gradient of the smoothed function. - For the analysis of Alg 2, the gossip matrix is assumed to be PSD. It may not be necessary, see "Can decentralized algorithms outperform centralized algorithms? a case study for decentralized parallel stochastic gradient descent, NIPS, 2017". - Line 212, the constraint is equal to \Theta * A = 0, that is not correct. The definition of A should be not be AA' = W. It should be AA' = W - I or others.